# Enhanced Bactericidal Efficacy of NaOCl at pH 12 Followed by Acidified NaOCl at pH 6.5 on *Enterococcus faecalis* Biofilm

## Ronald Wigler *, Shlomo Matalon, Tomer Goldberger, Anat Or Lerner and Anda Kfir

Department of Endodontology, The Goldschleger School of Dental Medicine, Tel Aviv University,
Tel Aviv 6934228, Israel; matalons@tauex.tau.ac.il (S.M.); tomergoldber@post.tau.ac.il (T.G.);
anatorlerner@tauex.tau.ac.il (A.O.L.); andak@post.tau.ac.il (A.K.)
* Correspondence: ronniewigler@tauex.tau.ac.il

**Abstract:** This study aimed to determine the bactericidal efficacy of sequential use of NaOCl pH 12 followed by acidified NaOCl pH 6.5, and compare it to that of either of these NaOCl solutions alone. *E. faecalis* biofilm was grown on standardized dentine specimens for four weeks. The specimens were randomly divided into four groups: (A) 4 min exposure to 0.9% saline solution (control); (B) 4 min exposure to 4% NaOCl pH 12; (C) 4 min exposure to 4% NaOCl pH 6.5; and (D) 2 min exposure to 4% NaOCl pH 12 followed by 2 min exposure to 4% NaOCl pH 6.5. The bactericidal activity was evaluated after the 4 min of contact time using confocal laser scanning microscopy. The volume ratio of red fluorescence to green and red fluorescence indicated the proportion of dead cells in the biofilm. The percent of dead cells in the saline solution group was significantly lower than those in the other groups. There was no significant difference between NaOCl pH 12 compared to NaOCl pH 6.5. The sequential use of NaOCl pH 12 followed by pH 6.5 significantly increased the percent of dead cells compared to both the samples exposed to either NaOCl pH 12 or pH 6.5. These results show that sequential irrigation protocol had a stronger bactericidal effect than the commonly used NaOCl pH 12.

**Keywords:** bactericidal efficacy; biofilm; confocal laser scanning microscopy; *Enterococcus faecalis*; NaOCl; pH

## 1. Introduction

Endodontic diseases involve microbial infections organized in biofilm structures [1]. These microbial communities are embedded in self-produced extracellular polymeric substances (EPSs) that attach to surfaces and form three-dimensional matrix structures, making the bacteria more resistant to antimicrobial agents [1–5]. Therefore, an effective root canal disinfection procedure requires the use of an irrigation solution with the ability to disrupt the biofilm matrix along with having a broad antimicrobial spectrum [5].

Sodium hypochlorite (NaOCl) is the most widely used irrigation solution in endodontics [6], and its antibacterial and tissue dissolution effectiveness depends on its pH, among other factors [7–9]. At pH 12, the regular type of NaOCl solution, the hypochlorite ion (OCl−) predominates in solution. At this pH, tissue dissolution of NaOCl is maximized and it has the capability to disrupt and reduce the matrix of biofilm-related matter [8–12].

On the other hand, when NaOCl is acidified to pH 6.5, hypochlorous acid (HOCl) is mostly present [12–14], and the bactericidal ability of NaOCl is maximized [9,10,12,14,15].

This study was designed with the rational of sequential use of NaOCl at two different pH: first disrupting the protective biofilm matrix, using NaOCl at pH 12, in order to allow subsequent better penetration of the more bactericidal acidified NaOCl (pH 6.5).

The aim of this study was to determine the bactericidal efficacy of sequential use of NaOCl at pH 12 followed by acidified NaOCl at pH 6.5, as compared to either NaOCl at pH 12 or NaOCl at

pH 6.5 alone. The null hypothesis was that there will be no difference between such sequential use and the bactericidal effect of NaOCl at either of the above pH levels alone.

## 2. Materials and Methods

The study design was approved by the Institutional Ethics Committee (IEC No. 132.16). The design was based on the study of Ruiz-Linares et al., in which bacterial biofilms that were grown on flat dentin slabs, were exposed to disinfecting solutions for 3 min, then subjected to confocal laser scanning microscopy, using live/dead staining [1]. Power analysis was conducted to achieve 85% power at a 0.05 significance level. According to the analysis, 5 specimens in each group would achieve 87% power.

### 2.1. Dentine Specimen Preparation

Intact human maxillary incisors with mature apices were selected from a pool of recently extracted teeth that were kept at 4 °C. Twelve teeth were horizontally sectioned at 1 mm and 5 mm apical to the cemento-enamel junction and then vertically sectioned along the mid-sagittal plane into 2 halves using a diamond-coated disc (Horico, Berlin, Germany) under water cooling. The outer cementum of each half was removed, and the inner part of the dentine root was ground and polished to create a flat surface. The size of the 24 samples was then adjusted, using a caliper, to obtain $4 \times 4$ mm (width × length) specimens. The smear layer formed during preparation of the specimens was removed by immersion in 4% NaOCl followed by 17% ethylenediaminetetraacetic acid (EDTA), each for 4 min, in an ultrasonic bath. The specimens were then immersed in 10% sodium thiosulfate for 4 min to inactivate the residual NaOCl, washed with distilled water and sterilized in an autoclave for 20 min at 121 °C. The dentine sections were then incubated in sterile brain heart infusion (BHI) broth for 24 h at 37 °C to verify the absence of turbidity in the culture medium.

### 2.2. Bacterial Biofilm Formation

*E. faecalis* (ATCC 29212) was plated on BHI broth supplemented with 2% (wt/vol) agar (Becton-Dickinson, Sparks, NV, USA) and incubated anaerobically at 37 °C for 24 h. The dentine specimens were placed individually in separate wells in a sterile 24-well plate. In each well containing a dentine specimen 50 μL of overnight *E. faecalis* culture (107 CFU/mL) was added and dispersed in 450 μL of fresh BHI growth medium. The plate was incubated anaerobically for 4 weeks at 37 °C, and the BHI medium was replenished every other day.

The dentine specimens were removed from the wells aseptically and gently rinsed with 1 mL sterile BHI broth to remove the non-adherent bacteria and then transferred to new sterile wells. Four sample were used to verify the presence of a multi-layer bacterial biofilm, using confocal laser scanning microscope (CLSM).

### 2.3. Disinfection of Dentine Covered with Biofilm

The specimens were randomly allocated to 4 groups (n = 5), according to the irrigation solutions to which the biofilms were to be exposed: Group A: 0.9% saline solution, which served as control; Group B: 4% NaOCl at pH 12; Group C: acidified 4% NaOCl at pH 6.5; and Group D: sequential application of 4% NaOCl at pH 12, followed by acidified 4% NaOCl at pH 6.5.

To prepare acidified 4% NaOCl at pH 6.5, 10 mL of 4% NaOCl (Merck, Darmstadt, Germany) at pH 12 was mixed with 0.3 mL of 99.5% acetic acid (Merck, Darmstadt, Germany) and allowed to settle for 5 min before being used in the study [9]. The pH of the solutions was determined by using a digital pH meter (Hanna Instruments, Woonsocket, RI, USA).

A droplet of 50 μL of each irrigation solution was placed on the biofilm surface for 4 min in groups A–C. In group D sequential exposure to the solutions was used: first pH 12 NaOCl solution was placed on the biofilm surface for 2 min, then removed and pH 6.5 NaOCl was used for the next 2 min, with a total exposure time of 4 min (similar to groups A–C). A sterile micro pipette was used to gently remove the first solution droplet from the sample surface before adding the second one. The pipet tip was

directed to the edge of the sample, and the aspiration of the solution was done slowly, taking care not to disturb the biofilm. After the contact period (4 min), the NaOCl solutions were removed (as above) and remaining NaOCl was inactivated by adding 50 μL of 10% sodium thiosulfate for 5 min and then rinsed with 1 mL of 0.9% sterile saline solution.

*2.4. Disinfection Assessment*

The dentine specimens from each group were stained with fluorescent LIVE/DEAD BacLight Bacterial Viability Stain (Molecular Probes, Eugene, OR, USA) and observed with CLSM (Leica TCS SP5, Leica Microsystems, Mannheim, Germany). The respective absorption and emission wave lengths were 500/550 nm for the SYTO 9 stain and 580/630 nm for the PI stain. Sequential frame scan mode was used to prevent crosstalk.

Three microscopic confocal volumes from random areas were acquired from each specimen using a 40× oil lens with a 2 μm step size and a format of 512 × 512 pixels. Single-channel imaging and simultaneous dual-channel imaging were used to display green fluorescence (live cells) and red fluorescence (dead cells). CLSM images of the biofilms were analyzed and quantitated by using Imaris 8.4.3 microscopy image analysis software (Bitplane AG, Zurich, Switzerland). The volume ratio of red fluorescence to green and red fluorescence in the three-dimensional reconstructions indicated the percent of dead cells in the biofilm. The total volume of green and red fluorescence was used to compare the amount of biomass between the groups.

*2.5. Statistical Analysis*

Statistical analysis was performed using mixed model test to determine significant clusters among groups. The level of significance was $p < 0.05$. All statistical analyses were performed using SAS 9.4 software (SAS Institute, Cary, NC, USA).

## 3. Results

A total of 60 CLSM operative fields (3-dimensional stacks) were evaluated, 15 from each group. The mean, median, minimum, maximum and standard deviation of $\log_{10}$ total biovolume ($\mu m^3$) are presented in Table 1. The mean and range (box plots) of the percent of red cells (dead cells) are shown in Figure 1.

**Table 1.** Biovolume of the biofilm after exposure to the irrigants.

| Irrigant | $\log_{10}$ Total Biovolume ($\mu m^3$) | | | | |
|---|---|---|---|---|---|
| | Mean | SD | Min. | Max. | Median |
| Saline | 7.35 | 0.22 | 7.03 | 7.40 | 7.68 |
| NaOCl pH 12 | 7.28 | 0.23 | 6.94 | 7.36 | 7.74 |
| NaOCl pH 6.5 | 7.16 | 0.29 | 6.73 | 7.14 | 7.61 |
| NaOCl pH 12 + pH 6.5 | 7.32 | 0.22 | 7.03 | 7.36 | 7.62 |

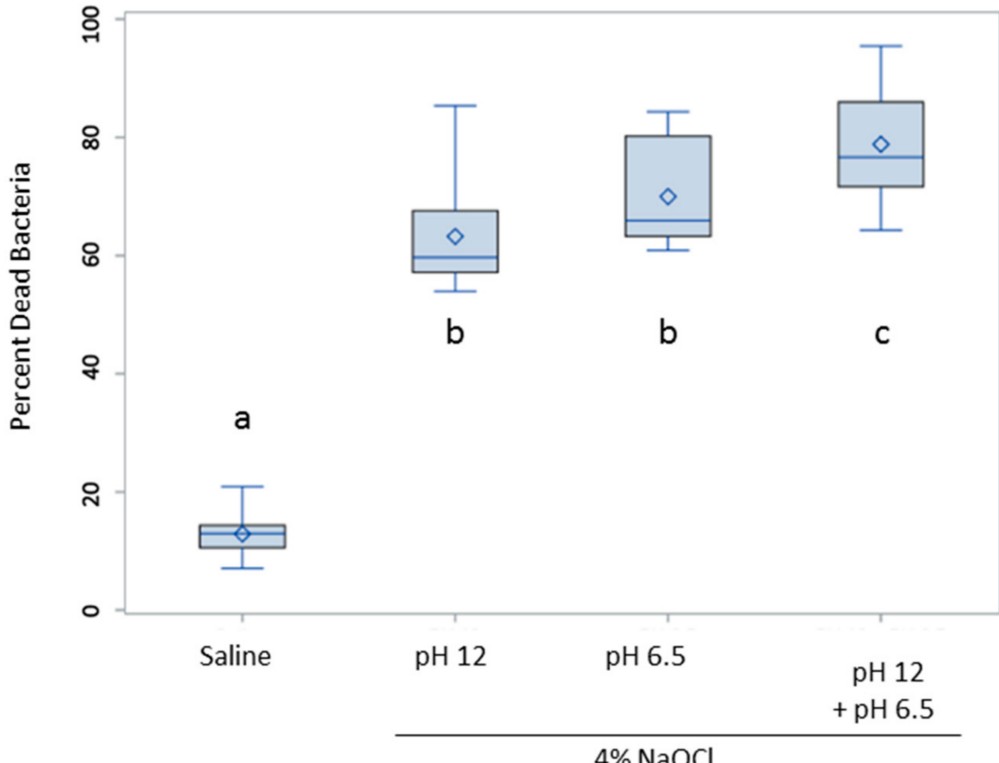

**Figure 1.** Mean and range (box plots) of the percent of red cells (dead cells) after 4 min of contact with the solutions. The bactericidal effect of the irrigants on *E. faecalis* biofilms is expressed as the ratio of red cells to green and red cells, indicating the proportion of dead cells in the biofilm. The percent of dead cells in the 0.9% saline solution group was significantly lower than those in the other groups ($p < 0.0001$). There was no significant difference in the percent of dead cells between 4% NaOCl pH 12 compared to acidified 4% NaOCl pH 6.5 ($p = 0.0802$). The sequential use of 4% NaOCl pH 12 followed by 4% NaOCl pH 6.5 significantly increased the percent of dead cells compared to both the samples exposed to either 4% NaOCl pH 12 or 4% NaOCl pH 6.5 ($p < 0.0008$ and $p < 0.0276$, respectively). Similar low case letters indicate no difference; different letters indicate significant difference between the groups. The box represents the interquartile range (IQR) and extends from Q1 to Q3. The rhombus indicates the mean and the horizontal bar indicates the median. Upper and lower "wiskers" indicate the maximum and minimum values.

For the total biovolume, there was no significant difference between the groups ($p = 0.2786$). For percentage of dead cells, the biofilms exposed to 0.9% saline solution had a significantly lower percent (13% ± 4%) than the other groups ($p < 0.0001$). There was no significant difference in the percent of dead cells between 4% NaOCl at pH 12 (63% ± 9%) compared to 4% NaOCl at pH 6.5 (70% ± 9%) ($p = 0.0802$). Sequential exposure of the biofilms to 4% NaOCl pH 12 followed by 4% NaOCl pH 6.5 significantly increased the percent of dead cells (79% ± 9%) compared to both the samples exposed to either 4% NaOCl pH 12 or 4% NaOCl pH 6.5 ($p < 0.0008$ and $p < 0.0276$, respectively). Representative CLSM images of the treated biofilms can be found in Figure 2.

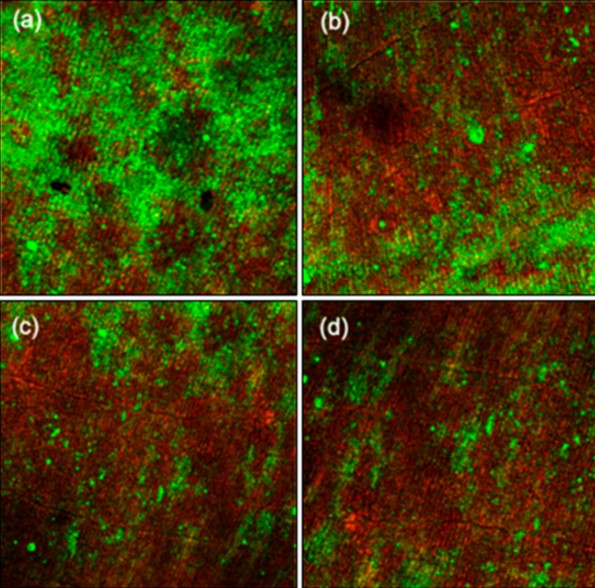

**Figure 2.** Representative confocal laser scanning microscopy images. (**a**) After exposure to saline (control); (**b**) after exposure to 4% NaOCl pH 12; (**c**) after exposure to acidified 4% NaOCl pH 6.5; and (**d**) after exposure to 4% NaOCl pH 12, followed by acidified 4% NaOCl at pH 6.5.

## 4. Discussion

The goal of endodontic treatment is to prevent or eliminate apical pathosis by means of shaping, cleaning, disinfecting, and sealing of the root canal system. Studies have proven that mechanical cleaning alone is limited in its ability to render the canal walls free from biofilm, since at least 10% of the walls remain untouched following instrumentation with the most modern rotary nickel-titanium file systems [16,17]. Thus, reduction of the microbial load and disruption of biofilms are commonly achieved by a combination of mechanical instrumentation and irrigation with tissue dissolving and antimicrobial irrigants [18].

NaOCl is currently the most widely used irrigation solution due to its antibacterial and tissue-dissolving properties [6,18]. It has been previously shown that these properties are affected by the pH of the NaOCl solution [7–9,15]. Various concentrations of NaOCl are being used clinically, ranging from 0.5–5.25% and above, with pH levels of 12 and above [6,18]. Solutions with lower pH levels are not commercially available since buffering NaOCl renders the solution unstable with a decrease in its shelf life [6,9,15]. In the present study, acidified NaOCl was prepared by adding acetic acid, according to the protocol of Mercade et al. [9]. Acetic acid is a suitable buffer solution since it has no effect on the available chlorine [8]. Nevertheless, acidification of NaOCl should be done carefully, as excessive pH reduction may negatively affect the dentin.

NaOCl at pH 12 has the capability to reduce biofilm-related matter [11], break glycosidic bonds, dissolve glycoconjugates in the biofilm matrix and cause lysis to bacterial cells [11].

Ruiz-Linares et al. [1], used a similar model to the one used in the present study and reported that 2.5% NaOCl substantially reduced the total biovolume of the biofilm compared to biofilms exposed to distilled water. This result was attributed to the disruption of the structure of the biofilm. In the present study, no such reduction of biovolume was found, and the total biovolume of the biofilms exposed to either of the NaOCl irrigants was not different from the biofilm exposed to saline. A potential explanation could be a difference in the bacteria and conditions used to grow the biofilm. In the former study, microbial samples were collected from infected root canals, and the polymicrobial biofilms (of undefined nature) were grown for three weeks with a weekly refreshment of the BHI medium. It could be that the *E. faecalis* biofilm grown, in the present study, for four weeks while replenishing the BHI medium every two days created a biofilm more resistant to disruption.

Additionally, based on previous studies, one could expect to find a more effective reduction in the total biovolume after treatment with NaOCl at pH 12 compared to the acidified irrigant at pH 6.5 [9–12,14]. Such a difference was not found in the present study. The above potential explanation may be in line with previous findings that *E. faecalis* is a relatively resistant bacterium [19,20]. Nevertheless, this issue warrants further investigation.

*E. faecalis* was chosen for the present study because it has the ability to survive in vivo as a single organism without the support of other bacteria and because it has been extensively used in antimicrobial endodontic studies [9,19,21,22]. The benefit of the simplified in vitro system used in the present study is its ability to isolate and study a single factor (pH of the NaOCl solutions) and avoiding many confounding factors that could result from variations in the biofilm constituents if a mixed biofilm of unidentified nature was used.

The bactericidal ability of the acidified NaOCl solution alone in the present study was not in agreement with previous reports, which indicated that the bacteria killing was maximized at lower pH [6,9,15]. Mercade et al. [9], proved that the antibacterial activity of a 4.2% NaOCl solution is enhanced by weak acidification of the solution to pH 6.5. In the present study, when used alone, NaOCl at pH 6.5 showed apparent enhanced bactericidal activity compared to NaOCl at pH 12, with 70% and 63% dead *E. faecalis* cells, respectively, but with no significant difference was found by the effects of the two solutions. When these two solutions were used in succession, one after the other, a significantly higher bactericidal effect was found (79% dead *E. faecalis* cells) compared to each solution alone. None of the tested solutions were able to completely eradicate the biofilm.

It may be assumed that the first NaOCl solution at pH 12 had a disruptive effect on the biofilm matrix [11,14], which allowed the acidified solution (pH 6.5), with its enhanced bactericidal activity [12,14], better access to the bacteria in the biofilm, thus enhancing the total bactericidal effect. Nevertheless, this assumption should be further investigated.

In the current study the biofilm was grown on flat dentin surface, similarly to Ruiz-Linares et al. [1]. This was done to allow optimal observation conditions for the confocal laser scanning microscopy and avoid potential distortions that may result from growing the biofilm on a concave dentin surface of a root canal [23]. The exposure time to the NaOCl solutions in the present study (4 min) was similar to that used in previous studies: Rôças et al. [21], (3 min) and Ordinola-Zapata et al. [24], (5 min).

The results of the present study show a promising direction in the use of NaOCl. It should be noted that this is a laboratory study and hence its limitations. We did not attempt to mimic the clinical situation. Additional studies using tooth models, a multi-species biofilm model and different exposure times are needed to confirm the promising results observed in the current study.

## 5. Conclusions

Within the limitations of the study, it can be stated that sequential exposure of *E. faecalis* biofilm to NaOCl at pH 12 followed by acidified NaOCl at pH 6.5 may have a stronger bactericidal effect on *E. faecalis* biofilm compared to the currently commonly used irrigation protocol using NaOCl at pH 12.

**Author Contributions:** Conceptualization, Investigation, Methodology, Writing-original draft, Writing-review & editing, R.W.; Investigation, Methodology, Resources, S.M.; Investigation, T.G.; Investigation, Methodology, Resources, A.O.L.; Investigation, Methodology, Writing-original draft, Writing-review & editing, A.K. All authors have read and agreed to the published version of the manuscript.

**Funding:** This research received no external funding.

**Conflicts of Interest:** The authors declare no conflict of interest.

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
