# Peer review of "Enhanced Bactericidal Efficacy of NaOCl at pH 12 Followed by Acidified NaOCl at pH 6.5 on Enterococcus faecalis Biofilm"

_applsci, doi:10.3390/app10176096_

Round 1

Reviewer 1 Report

This paper evaluates the bactericidal efficacy of different NaOCl solutions against the E. Faecalis growth on standardized dentin specimens, in vitro.
The specimens were randomly divided into 4 groups: (A) 0.9% saline solution (control), (B) 4% NaOCl pH 12, (C) 4% 14 NaOCl pH 6.5, and (D) 4% NaOCl pH 12 followed by 4% NaOCl pH 6.5.
In the abstract, the autors declare that the protocol (D), which is a double rinse of solution is more effective than the other ones. The results seems to be obvious since a double rinse of disinfectant resulted more efficient than a single one. It's of common use to make different rinses of disinfectant during endodontic procedures due to make the treatment more efficient.
This was also confirmed by the sentence: "There was no significant difference between NaOCl pH 12 compared to NaOCl pH 6.5.", which confirms that the two disinfectants are similar, but the double rinse is more efficient than a single one.
However, in the materials and methods section this fact is clearified by the fact that the double rinse was performed by two rinses of 2 minutes, equal to the other ones. So I think that this aspect should be clarified also in the abstract.
The authors declare that the Ph 6.5 NaOCl solution is not available on the market since it's unstable and it should be prepared just before the procedure, so it represents a newly introduced difficulty for the clinician, which could prepare a wrong solution which could affect the dentin due to it's lower Ph. This aspect should be emphatized.
In the abstract is it possible to change "...it to that of either of these..." in a more clean sentence?
Please remove double spaces.
at line 141 a "." is missing at the end of the sentence.
No spaces should be present while representing standard deviations: e.g. "(13% ±4%)".
The overall paper is well written and the experimental conditions are very clearly exposed. The english is correct and only some typos should be fixed.
References do not respect journal's format so they should be checked and corrected.

Author Response

Reviewer 2

Comments and Suggestions for Authors

This paper evaluates the bactericidal efficacy of different NaOCl solutions against the E. Faecalis growth on standardized dentin specimens, in vitro.
The specimens were randomly divided into 4 groups: (A) 0.9% saline solution (control), (B) 4% NaOCl pH 12, (C) 4% 14 NaOCl pH 6.5, and (D) 4% NaOCl pH 12 followed by 4% NaOCl pH 6.5.

1. Comment:

In the abstract, the autors declare that the protocol (D), which is a double rinse of solution is more effective than the other ones. The results seems to be obvious since a double rinse of disinfectant resulted more efficient than a single one.

Response:

The issue was addressed and clarified in the Abstract.

2. Comment:

It's of common use to make different rinses of disinfectant during endodontic procedures due to make the treatment more efficient.

Response:

We agree and we are searching for the best solutions and combinations.

3. Comment:

This was also confirmed by the sentence: "There was no significant difference between NaOCl pH 12 compared to NaOCl pH 6.5.", which confirms that the two disinfectants are similar, but the double rinse is more efficient than a single one.
However, in the materials and methods section this fact is clearified by the fact that the double rinse was performed by two rinses of 2 minutes, equal to the other ones. So I think that this aspect should be clarified also in the abstract.

Response:

The issue was clarified in the Abstract.

4. Comment:

The authors declare that the pH 6.5 NaOCl solution is not available on the market since it's unstable and it should be prepared just before the procedure, so it represents a newly introduced difficulty for the clinician, which could prepare a wrong solution which could affect the dentin due to it's lower Ph. This aspect should be emphatized.

Response:

The point was addressed and emphasized (Discussion, line ……)

5. Comment:

In the abstract is it possible to change "...it to that of either of these..." in a more clean sentence?

Response:

The sentence was amended to be better understood.

6. Comment:

Please remove double spaces.

Response:

Double spaces were removed.

7. Comment:

at line 141 a "." is missing at the end of the sentence.

Response:

Corrected: the "." was added in current line 142.

8. Comment:

No spaces should be present while representing standard deviations: e.g. "(13% ±4%)".

Response:

Corrected, as suggested.

9. Comment:

The overall paper is well written and the experimental conditions are very clearly exposed.

Response:

We thank the reviewer for the comment.

10. Comment:

The english is correct and only some typos should be fixed.

Response:

Typos were fixed, as suggested.

Comment:

References do not respect journal's format so they should be checked and corrected.

Response:

References were corrected to the Journal’s format.

Reviewer 2 Report

Scientific species names as E. faecalis must be in italics.

figure 1

statistics should be included into the graph;

furthermore the box plots have to be more described. What does the line, the diamond symbol and the error bars represent?

line 115: ".. log10.." subcript the 10

line 141: " ... Figure 1" - correct to figure 2

Author Response

Reviewer 1

1. Comment:

Scientific species names as E. faecalis must be in italics.

Response:

faecalis was italized.

2. Comment:

figure 1 statistics should be included into the graph;

Response:

Statistics were included in the graph.

3. Comment:

furthermore the box plots have to be more described. What does the line, the diamond symbol and the error bars represent?

Response:

The details of the box-plot were explained in the figure legend.

4. Comment:

line 115: ".. log10.." subcript the 10

Response:

Corrected Log 10 was subscripted.

5. Comment:

line 141: " ... Figure 1" - correct to figure 2

Response:

Corrected: Figure 1 was corrected to figure 2